# Armed Phages: A New Weapon in the Battle Against Antimicrobial Resistance

**DOI:** 10.3390/v17070911

**Published:** 2025-06-27

**Authors:** Cleo Anastassopoulou, Deny Tsakri, Antonios-Periklis Panagiotopoulos, Chrysa Saldari, Antonia P. Sagona, Athanasios Tsakris

**Affiliations:** 1Department of Microbiology, Medical School, National and Kapodistrian University of Athens, 11527 Athens, Greece; cleoa@med.uoa.gr (C.A.); dtsakri@med.uoa.gr (D.T.); anpanagiotop@med.uoa.gr (A.-P.P.); chrysasaldari@gmail.com (C.S.); 2School of Life Sciences, University of Warwick, Coventry CV4 7AL, UK; a.sagona@warwick.ac.uk

**Keywords:** bacteriophages, phage therapy, antimicrobial resistance, bacterial defenses, genetically engineered phages, CRISPR-Cas, NAD+ depletion

## Abstract

The increasing prevalence of multidrug-resistant (MDR) bacterial infections necessitates the exploration of alternative antimicrobial strategies, with phage therapy emerging as a viable option. However, the effectiveness of naturally occurring phages can be significantly limited by bacterial defense systems that include adsorption blocking, restriction–modification, CRISPR-Cas immunity, abortive infection, and NAD+ depletion defense systems. This review examines these bacterial defenses and their implications for phage therapy, while highlighting the potential of phages’ bioengineering to overcome these barriers. By leveraging synthetic biology, genetically engineered phages can be tailored to evade bacterial immunity through such modifications as receptor-binding protein engineering, anti-CRISPR gene incorporation, methylation pattern alterations, and enzymatic degradation of bacterial protective barriers. “Armed phages”, enhanced with antimicrobial peptides, CRISPR-based genome-editing tools, or immune-modulating factors, offer a novel therapeutic avenue. Clinical trials of bioengineered phages, currently SNIPR001 and LBP-EC01, showcase their potential to safely and effectively combat MDR infections. SNIPR001 has completed a Phase I clinical trial evaluating safety in healthy volunteers, while LBP-EC01 is in Phase II trials assessing its performance in the treatment of *Escherichia coli*-induced urinary tract infections in patients with a history of drug-resistant infections. As “armed phages” progress toward clinical application, they hold great promise for precision-targeted antimicrobial therapies and represent a critical innovation in addressing the global antibiotic resistance crisis.

## 1. Introduction

The use of antibiotics revolutionized medicine, drastically improving patient outcomes and saving millions of lives [1,2]. However, the widespread misuse and overprescription of these drugs created intense selective pressure on bacteria, fostering the emergence of multidrug-resistant (MDR) strains that no longer respond to conventional treatments [1,3,4]. As a result, antimicrobial resistance (AMR) has become a major global health threat. According to the World Health Organization (WHO), AMR was the direct cause of 1.27 million deaths worldwide in 2019, while additionally contributing 4.95 million fatalities [5]. Future projections indicate an even more alarming scenario, with an estimated 1.91 million deaths directly attributed to drug-resistant infections and 8.22 million AMR-related deaths anticipated by 2050 [6,7]. These figures underscore the urgent need to develop and implement effective strategies to combat the growing threat of antibiotic resistance. The rise in AMR, coupled with the lack of discovery of new antibiotics in over three decades, has compelled the scientific community to prioritize the development of alternative treatments to combat MDR bacteria [8,9]. One of these alternative treatments involves the resurgence of phage therapy, which was abandoned in Western Europe and the United States nearly 100 years ago, although it continued to be explored in certain Eastern European settings [3,10].

Phages are primarily categorized based on their morphology and nucleic acid composition, which can be either DNA or RNA in single-stranded (ss) or double-stranded (ds) forms [11]. In 2022, the International Committee on Taxonomy of Viruses (ICTV) formally abolished the morphology-based families *Myoviridae*, *Siphoviridae*, and *Podoviridae*, along with the order *Caudovirales*, due to their polyphyletic nature [12,13]. These tailed dsDNA viruses are now grouped under the class *Caudoviricetes*, which encompasses newly defined families and genera based on genomic similarity rather than virion morphology. Among the newly established taxa, the order *Crassvirales* and several families infecting *Flavobacteriia* highlight the increasing reliance on metagenomic datasets to define evolutionarily coherent virus groups [12]. Structurally, phages consist of a proteinaceous capsid that coats their genetic material [14]. The design of their capsids varies significantly, influencing the mode of infection of host bacteria [15]. Phage tails often function like a syringe, enabling them to penetrate bacterial cell walls and membranes to deliver their genetic material directly into the host [16].

Phages follow two main life cycles: the lytic cycle and the lysogenic cycle. In the lytic cycle, a phage attaches to a bacterial cell, injects its genetic material, and hijacks the host’s cellular machinery to produce new phage particles [17]. This process ultimately causes the bacterial cell to rupture, releasing newly formed phages that then go on to infect other bacteria [17]. A well-known example of a virulent phage that exclusively follows this cycle is bacteriophage T4, which infects *Escherichia coli* [18]. The lysogenic cycle involves the integration of the phage genome into the bacterial host’s DNA, allowing it to remain dormant and replicate with each bacterial cell division [19,20]. Under certain stressful microenvironmental conditions, lysogeny can be disrupted, with the switch to the lytic cycle leading to the production of new phages and the eventual lysis of the host cell [19]. A classic example of a temperate phage capable of lysogeny is phage lambda, which also infects *E. coli* [21]. Phages that exclusively undergo the lytic cycle are classified as virulent, whereas temperate phages have the ability to alternate between the lytic and lysogenic cycles [22].

Given the alarming rise in drug-resistant bacteria and the stagnation in new antibiotic development, alternative antimicrobial strategies such as phage therapy have re-emerged as promising tools in the fight against MDR infections [23]. The effectiveness of phage therapy may, nonetheless, be hindered by bacterial defense systems, which have evolved to block phages’ infection through diverse molecular mechanisms [24]. These barriers highlight a critical challenge in the clinical application of natural phages. This review examines key bacterial defense strategies and their impact on phage therapy, focusing on the innovative ways in which phage bioengineering is used to overcome these challenges as “armed phages” enter the clinic.

## 2. Evolutionary Dynamics of Phages-Bacteria Interactions

### 2.1. Mechanisms of Phages’ Infection of Bacterial Cells

Phage infection begins with adsorption, where phages attach to bacterial hosts via interactions between viral receptor-binding proteins and bacterial surface receptors [25]. This attachment is achieved through specific interactions between phage receptor-binding proteins and bacterial surface receptors [26]. This occurs in three stages: initial contact, a reversible interaction that keeps the phages near the host, and an irreversible attachment to a secondary receptor, securing their position for infection [25]. Phage-derived enzymes, depolymerases, can aid adsorption by degrading capsular polysaccharides that block receptors on the bacterial surface [27]. After attachment, phage-encoded virion-associated lysins (ectolysins) degrade the peptidoglycan layer, creating an entry point for the phage genome and initiating infection [27]. Conformational changes in the tail machinery then facilitate genome injection into the bacterial cytoplasm, initiating infection [28].

The phage hijacks the host’s molecular machinery for replication, using its RNA polymerase, ribosomes, and metabolic energy to transcribe and translate the phage genome [29]. Early genes often encode proteins that suppress host gene expression, redirect RNA polymerase specificity, and initiate viral DNA replication [30]. Middle and late genes encode structural proteins and lytic enzymes required for virion assembly and host lysis [30,31]. During replication, multiple copies of the viral genome are generated, and structural components self-assemble to form new virions. Εnzymatic degradation of the bacterial peptidoglycan layer, a crucial step for host lysis, follows [32]. Holins accumulate in the inner membrane, forming pores that allow endolysins to degrade peptidoglycan [32]. In Gram-negative bacteria, spanins destabilize the outer membrane, enabling virion escape [33]. The lytic burst releases dozens to hundreds of new phages, continuing the infection cycle.

### 2.2. Bacterial Defense Strategies Against Phages

Understanding the mechanisms by which phages infect and subjugate bacterial cells provides crucial insight into the dynamic interactions between these viruses and their hosts. While phages have evolved highly specialized strategies to recognize, penetrate, and replicate within bacterial cells, bacteria are not passive targets. They have developed a diverse array of defense mechanisms to counteract phage infections and ensure survival [34]. These bacterial defense systems, ranging from surface modifications that prevent phage adsorption to sophisticated intracellular immunity pathways, reflect the ongoing evolutionary arms race between bacteria and phages, leading to the development of resistance and counter-resistance strategies [34]. The next section explores these bacterial defense mechanisms, highlighting the plurality of employed molecular strategies.

#### 2.2.1. Adsorption Blocking

To counter adsorption, bacteria have evolved multiple defense mechanisms that obstruct phage attachment by modifying or concealing receptors or by creating physical barriers. One common strategy is receptor modification, where bacteria mutate genes responsible for surface receptor synthesis, altering their structure or reducing receptor availability [35]. Some bacteria, such as *Klebsiella pneumoniae*, modify their capsular polysaccharides (CPSs) to prevent phage recognition [36]. Other bacterial species can produce alternative proteins that mask receptors, thereby interfering with phage attachment [35].

Another major strategy employed by bacteria involves extracellular polymer production, where bacteria secrete exopolysaccharides and other matrix components to form a protective barrier [37]. This extracellular matrix not only provides structural and physiological benefits to bacteria but also serves as a physical defense against phage infection. For instance, *Azotobacter chroococcum* produces sodium alginate, a polymer that reinforces the bacterial surface, significantly reducing phage adsorption [38]. By embedding themselves in these extracellular structures, bacteria can limit phage penetration, effectively reducing their susceptibility to viral infection. Some bacteria can even regulate the production of these protective barriers in response to phage presence, further complicating therapeutic interventions [39].

#### 2.2.2. Nucleic Acid Protection

##### Restriction–Modification Systems

One of the best-characterized and abundant bacterial defense mechanisms against phage infection is the restriction–modification (R-M) system [40]. R-M systems play a crucial role in protecting the host by distinguishing between self and foreign DNA [40]. The defensive function of R-M systems relies on two key enzymatic components: restriction endonucleases (REases) and methyltransferases (MTases) [41]. MTases selectively methylate specific recognition sites on the host’s genomic DNA, marking it as self and preventing degradation [41]. In contrast, REases identify and cleave unmethylated foreign DNA, such as that introduced by phages, effectively neutralizing the threat before it can establish infection [41].

R-M systems are classified into four major types based on the composition and function of their enzymatic components [42]. Type I R-M systems consist of a multi-subunit protein complex, including an MTase, REase, and a specificity (S) subunit, with the S subunit determining the DNA sequences targeted for both methylation and restriction, ensuring precise recognition [43]. Type II R-M systems feature independent MTase and REase enzymes that function separately, making them the primary source of commercially available restriction enzymes used in molecular cloning [42,44]. Type III R-M systems also contain distinct MTase and REase proteins, but these operate cooperatively rather than independently [45]. Type IV R-M systems lack a methyltransferase altogether and specifically degrade foreign DNA that has been chemically modified, a strategy believed to have evolved to counter phages that evade restriction by altering their genetic material [46]. Notably, MTases tend to be more evolutionarily conserved than REases, as restriction enzymes undergo rapid adaptation to counteract phage mutations [42].

Phages, on the other hand, have evolved various mechanisms to bypass bacterial R–M systems, ensuring their survival and replication. One strategy involves minimizing the number of recognition sites in their genome or spacing them in a way that prevents effective cleavage by the bacterial host’s restriction enzymes [47]. For instance, wild-type *Staphylococcus aureus* K-like phages exhibit a broad host range, partly because they lack recognition sites for many R-M systems, demonstrating how such genomic features contribute to the evasion of defense mechanisms of different bacterial strains [48]. Another approach is exploiting the host’s own methylation system, allowing the phage DNA to be modified and protected from enzymatic degradation [47]. Some phages also introduce proteins alongside their genome that shield restriction sites, preventing bacterial enzymes from targeting them [47]. Other phages produce proteins that mimic DNA, effectively binding and neutralizing restriction enzymes to inhibit their function [47].

##### The CRISPR-Cas System

Apart from R-M systems, bacteria can cleave invading phage genomes through the action of the CRISPR (clustered regularly interspaced short palindromic repeat)—Cas (CRISPR-associated) system [49]. This system consists of two key components: a CRISPR array and Cas proteins [49]. CRISPR array functions as a molecular archive, storing genetic fragments (spacers) from previously encountered phages between repetitive DNA sequences, creating a reference for future recognition [50]. Cas proteins, encoded by nearby *cas* genes, play a crucial role in detecting and neutralizing foreign genetic material [50].

The CRISPR-Cas system defends against phages through three distinct stages: adaptation, expression, and interference [51]. During adaptation, bacteria integrate short fragments of foreign DNA from invading phages into their CRISPR array, forming a genetic record of past infections [51]. In the expression stage, these stored sequences are transcribed into small CRISPR RNAs (crRNAs), which associate with Cas proteins to form an RNA-guided surveillance complex [51]. During interference, the crRNA directs the Cas nucleases to recognize and degrade matching foreign genetic material upon subsequent phage infections, preventing the virus from establishing an infection [52].

The CRISPR-Cas system is classified into two broad categories based on structural and functional characteristics [53]. Class 1 systems, which include types I, III, and IV, rely on multi-subunit protein complexes for defense, whereas Class 2 systems, comprising types II, V, and VI, utilize a single-protein effector for interference [53]. While both classes provide adaptive protection against phages, Class 1 systems are more widespread and often employ a highly efficient DNA degradation mechanism [53].

Phages have developed multiple strategies to evade bacterial CRISPR-Cas defenses. Initially, simple point mutations in the protospacer-adjacent motif (PAM) or seed regions were found to be sufficient to disrupt CRISPR-Cas targeting [54]. However, this strategy alone is often inadequate for long-term survival due to the adaptive nature of the bacterial response [55]. To further escape detection, some phages chemically modify their DNA, e.g., through hydroxymethylation, which helps them avoid degradation [55]. More significantly, phages have evolved dedicated protein inhibitors known as anti-CRISPRs, which actively suppress CRISPR-Cas function [56]. These anti-CRISPR proteins can interfere with the bacterial defense system by preventing DNA binding or inhibiting the cleavage activity of CRISPR-associated proteins [56]. For example, AcrIIA4 from *Listeria monocytogenes* binds to the Cas9-sgRNA complex and inhibits its nuclease activity without preventing DNA binding [57]. In contrast, AcrF1, which targets the type I-F system in *Pseudomonas aeruginosa*, binds to the Csy surveillance complex and blocks base pairing between the crRNA and the target DNA, thereby preventing DNA recognition [58]. Since their discovery in 2013, over 90 families of anti-CRISPR proteins have been identified, primarily targeting type I and type II CRISPR-Cas systems [59].

#### 2.2.3. Abortive Infection

The abortive infection (Abi) system is a bacterial defense mechanism that prevents the spread of phage infections by triggering cell death or metabolic shutdown once an infection is detected [60]. Unlike other bacterial defense strategies that directly eliminate phage genetic material, Abi functions as a last-resort mechanism when early defenses have failed [60]. By sacrificing the infected cell, the system ensures that no mature phage particles are released, thereby protecting the surrounding bacterial population [60]. A well-characterized example is the Rex system, found in *E. coli* strains lysogenized by phage λ [61]. This system relies on RexA, a sensor that detects phage infection by recognizing phage protein-DNA complexes, and RexB, an ion channel embedded in the membrane [61]. When RexA is activated, it stimulates RexB, which disrupts the membrane potential, leading to a significant drop in ATP levels [61]. As a result, the synthesis of essential macromolecules is suppressed, preventing the bacterial cell from multiplying [61]. Because phage replication is also dependent on ATP and ATP-driven cellular processes, the infection is effectively aborted before viral particles can form [61]. Nevertheless, some phages have evolved strategies to bypass this defense mechanism [62]. In phage T4, for instance, the rIIA and rIIB proteins help suppress the effects of the Rex system [62].

#### 2.2.4. NAD+ Depletion Defense Systems

Nicotinamide adenine dinucleotide⁺ (NAD⁺) is a vital coenzyme found in all living cells, where it plays a central role in redox reactions, energy production, and metabolic regulation [63]. Its function is crucial for maintaining cellular homeostasis, and its availability is tightly regulated. Abrupt depletion of NAD⁺ can severely compromise cell viability [63]. Bacteria have evolved defense systems that exploit this vulnerability as a strategy to combat phage infection [63]. A key component of these systems is proteins containing Toll/interleukin-1 receptor (TIR) domains [63]. Depending on the defense system, TIR-domain proteins recognize different structural components of infecting phages, such as capsid, tail, or portal proteins, which activate their enzymatic function [64]. TIR domains respond to phage infection by synthesizing nicotinamide (NAM), ADPR, and cyclic ADPR (cADPR), leading to NAD⁺ depletion and disrupting critical metabolic and DNA repair pathways, which triggers programmed cell death [64]. These signaling molecules also activate downstream effector proteins, including members of the silent information regulator 2 (SIR2) family, which enzymatically degrade NAD⁺, thereby amplifying the cellular response to phage infection [64]. In addition, some TIR domain-containing proteins act as NADases, hydrolyzing NAD⁺ and exhibiting both nucleosidase and cyclase activities [64]. One example is the staphylococcal Thoeris defense system, which includes two TIR domain-containing sensors, ThsB1 and ThsB2, and an effector protein, ThsA [65]. Upon recognizing phage capsid proteins, the TIR sensors form a complex with them, initiating NAD⁺ degradation through the assistance of the ThsA effector [65]. Different bacterial species encode diverse anti-phage defense systems, such as Cyclic-oligonucleotide-Based Anti-phage Signaling Systems (CBASS) and Pyrimidine cyclase system for anti-phage resistance (Pycsar), highlighting the evolutionary variety in mechanisms used to sense and respond to phage infection [64].

Recent findings have shown that some phages have evolved strategies to counteract NAD⁺ depletion-based bacterial defense systems. These phages encode specialized enzymatic pathways capable of restoring NAD⁺ from its breakdown products within infected cells [66]. One such pathway, known as NAD⁺ reconstitution pathway 1 (NARP1), involves two enzymes that sequentially convert ADPR into NAD⁺ by first phosphorylating it and then combining the intermediate with nicotinamide [66]. This mechanism enables phages to bypass several NAD⁺-targeting bacterial defenses, including Thoeris and SIR2-associated systems [66]. A second pathway, NARP2, achieves NAD⁺ synthesis using alternative metabolites, further expanding the phages’ capacity to evade host immunity [66]. These adaptations reflect a sophisticated viral countermeasure that neutralizes host-induced NAD⁺ depletion and supports continued phage replication.

## 3. Limitations of Naturally Occurring Phages in Therapeutic Applications

Phage therapy presents a promising alternative to antibiotics, particularly in addressing the growing AMR challenge [67]. However, naturally occurring phages have several limitations that hinder their effectiveness in clinical applications [67]. One of the primary challenges is their narrow host range, which necessitates precise identification of the bacterial strain and the availability of suitable phages for immediate use [68]. Additionally, the human immune response may not always be favorable, since it can potentially limit the ability of therapeutic phages to reach their target bacteria effectively [67]. Most critically, bacterial defense systems pose significant barriers to phage therapy by actively preventing phage infection and replication [69].

For example, phages that rely on specific surface receptors can become useless against bacteria that modify or hide these structures, thereby preventing infection [70]. Likewise, extracellular barriers can prevent phages from reaching their bacterial targets, significantly reducing therapeutic efficacy. Beyond physical barriers, other bacterial defense mechanisms, such as the Abi system and NAD+ depletion systems, present a significant challenge to phage therapy by causing premature bacterial cell death before phage replication can occur. The early destruction of the bacterial host prevents phage self-amplification, thereby reducing the effectiveness of phage therapy. Since self-regulation is a key advantage of phage therapy, overcoming these bacterial defense strategies is crucial for its therapeutic potential [71]. The limitations imposed on phage therapy by bacterial defense mechanisms and potential solutions offered by the genetic engineering of phages are summarized in Table 1.

## 4. Bioengineered Phages for Enhanced Therapeutic Potential

Although phages naturally co-evolve with bacteria, this ongoing evolutionary arms race limits their long-term therapeutic success, making it necessary to explore other solutions [69]. Among the strategies discussed during the Centennial Celebration of Bacteriophage Research in Georgia in 2017 was the use of phage cocktails, which has been a prominent approach in phage therapy since the early 20th century [72]. While combining multiple phages can enhance treatment efficacy and reduce resistance development, this approach has practical limitations, including the finite number of phages that can be incorporated into a single formulation [72]. That being said, successful clinical applications have demonstrated that well-designed phage cocktails can be effective in treating complex infections [73,74]. Furthermore, bacterial resistance even to phage cocktails will inevitably emerge over time, but recent studies have shown that such resistance may come at a biological cost [72,75]. Phage-resistant bacteria may, thus, become more susceptible to antibiotics, a trade-off that opens new therapeutic possibilities, encompassing phage cocktail-antibiotic synergy [75]. Given these challenges, phage bioengineering has emerged as a powerful complementary strategy enabling the enhancement of natural phage properties and introducing novel functionalities [70].

Examples of how the genetic manipulation of phages can counteract the defense mechanisms of bacteria include the following paradigms: (1) Modification of phage receptor-binding proteins to enable the recognition of multiple bacterial receptors, thus expanding the host range of bioengineered phages or enabling them to recognize and use multiple receptors on a bacterial host [76,77]. For instance, synthetic biology approaches have been used to reprogram phage host specificity by swapping tail components, enabling *E. coli* phage scaffolds to efficiently target *Klebsiella* and *Yersinia*, and vice versa, allowing the selective elimination of pathogenic bacteria within complex microbial communities [78]. (2) Introduction of depolymerase enzymes to degrade bacterial polysaccharides, thus improving penetration through extracellular barriers [79]. By implementing these strategies, researchers can develop more resilient and effective phage therapies, reducing the effect of adsorption blocking of bacterial defense systems and improving clinical outcomes.

Furthermore, the incorporation of a methylation system that would modify the phage DNA to evade degradation by the bacterial restriction enzymes could help avoid R-M defenses [80]. Such a system has been discovered in human gut phages [80]. Moreover, the incorporation of a functional methyltransferase gene has already been efficiently achieved using the CRISPR-Cas system [81]. Additionally, the inclusion of anti-CRISPR (*Acr*) genes in phages could prevent the CRISPR-Cas system from recognizing and cleaving the phage genome [82].

A potential concern is that conjugative plasmids can acquire Acr proteins, which disable CRISPR defenses and allow antibiotic resistance genes to evade destruction and spread between bacteria [83]. This process facilitates the horizontal AMR transfer, enabling bacteria to acquire and retain traits that render them less susceptible to antibiotics [83]. Consequently, there is concern that the use of bioengineered phages with payloads of Acr proteins could unintentionally accelerate the spread of antibiotic resistance by promoting the survival and dissemination of resistance-carrying plasmids [83].

The altruistic early destruction of infected bacteria to save the rest of the population, which is mediated by the Abi defense system and the NAD+ depletion systems, prevents phage self-amplification. However, given the recent advancements in synthetic biology and phage bioengineering, it may be possible to overcome this barrier by designing phages that express proteins capable of suppressing the effects of Abi systems [84]. Furthermore, recent findings open the door to engineering phages with built-in resistance to NAD⁺ depletion-based bacterial defenses. By incorporating NAD⁺ reconstitution pathways such as NARP1 or NARP2 into synthetic phage genomes, it may be possible to enhance phage survival and replication in the presence of TIR- and SIR2-mediated host immunity systems. Tailoring these proteins to specifically target the defense mechanisms of different bacterial hosts could enhance the success of phage therapy, making it a more viable treatment option against resistant bacterial infections [84].

## 5. “Armed Phages”: The Next Generation of Phage Therapy

### 5.1. Genetically Engineering Phage Genomes for Enhanced Therapeutic Results

“Armed phages” are phages that have been genetically engineered to carry additional functionalities to enhance their therapeutic performance [85]. These modified phages are created to transfer specific molecules directly to bacterial targets, aiming to treat infections, particularly those caused by MDR bacteria [86]. “Armed phages” can be classified according to their functionality (Figure 1).

It is important to note that armed phages may not necessarily be genetically engineered; natural phages manipulated to carry additional moieties, such as photosensitizers or antimicrobial peptides, are also considered to be “armed.” One of the most extensively studied approaches involves CRISPR—Cas equipped phages, which enable the precise targeting and degradation of specific bacterial DNA or plasmids, offering a highly specific antimicrobial strategy [87]. Another important category includes phages armed with enzymes capable of disrupting bacterial biofilms [88]. Phages with nanoparticles (NPs), incorporating materials such as silver or quantum dots, have also been developed to augment bactericidal activity [89,90]. Lastly, phages can be armed with photosensitizers, a strategy that generates reactive oxygen species (ROS) upon light activation to eliminate bacterial pathogens [91,92].

Phages can also be genetically modified to enhance their antimicrobial activity and their synergy with antibiotics [93]. “Armed phages” have been shown to boost the bactericidal effects of antibiotics like gentamicin and ampicillin, even against resistant strains, while also targeting biofilms [93]. To combat antibiotic resistance, temperate phages have been engineered to deliver antibiotic sensitivity genes (*rpsL* or *gyrA*), restoring bacterial susceptibility to streptomycin and nalidixic acid [94]. Other modifications enabled phages to interfere with quorum sensing, further disrupting bacterial communication and resistance mechanisms [93]. Phage-derived endolysins have also been optimized for improved antibacterial activity, as seen in the chimeric enzyme Cpl-711, which exhibited superior bactericidal efficacy against *Streptococcus pneumoniae* [95]. These strategies highlight the potential of phage engineering to enhance antibiotic efficacy, combat resistance, and improve antimicrobial therapies.

### 5.2. Applications of the CRISPR-Cas System

#### 5.2.1. CRISPR-Cas Systems in Phage Therapy

CRISPR-Cas delivery in target bacteria can be achieved by both virulent and temperate phages [96]. CRISPR-Cas systems are advantageous for the bioengineering of phages using either the Single-Plasmid-Mediated Asynchronous Recombination (SPMAR) or Dual-Plasmid-Mediated Synchronous Recombination (DPMSR) method [97]. The CRISPR-Cas system has been widely adapted for precise genetic targeting, with different variants offering distinct functionalities. Commonly studied systems include types II-A (CRISPR-Cas9), I-E (CRISPR-Cas3), III-A (CRISPR-Cas10), V (CRISPR-Cas12a), and VI-A (CRISPR-Cas13a) [91].

CRISPR-Cas9 is the most commonly used system, creating double-strand breaks, and has demonstrated therapeutic potential, such as combating methicillin-resistant *Staphylococcus aureus* (MRSA) infections in vivo [98]. In contrast, the type I CRISPR-Cas complex employs Cas3 to introduce single-strand nicks followed by processive 5′ → 3′ DNA degradation [99]. Kiro et al. were the first to genetically engineer the *Escherichia coli* phage T7 with the application of the type I-E CRISPR-Cas system [100]. Using homologous recombination, the T7 phage genome was modified, and the CRISPR-Cas system targeted unedited phage genomes [100]. Subsequently, the intended recombinant phages were separated, rendering this CRISPR-Cas system applicable for modifying any phage genome beyond the type II CRISPR-Cas, which incorporates the versatile Cas9 protein [100]. CRISPR-Cas3 was successful in targeting *Clostridium difficile* both in vivo and in vitro [101].

Other variants, such as CRISPR-Cas12, specialize in targeting DNA, whereas CRISPR-Cas13 targets single-stranded RNA (ssRNA), making it a promising tool for RNA-guided gene silencing applications, including against RNA viruses in controlled settings [99]. However, CRISPR-Cas systems require synthetic delivery vehicles (e.g., plasmids or phage capsids) to enter target bacterial cells. Notably, engineered CRISPR-Cas13a systems have demonstrated sequence-specific bactericidal activity against antimicrobial-resistant *E. coli* and *S. aureus* when packaged into phage capsids [102]. CRISPR-Cas13, especially the LbuCas13a protein, enables efficient and accurate genome editing across a variety of phages. It has been successfully applied to multiple diverse *E. coli* phages, allowing precise modifications from single nucleotides to larger gene deletions with high success rates [103]. These diverse CRISPR systems expand the potential for genetic editing to antiviral applications [99].

#### 5.2.2. Clinical Applications of the CRISPR-Cas System

A strategy to delay the onset of phage resistance involves designing phages equipped with the CRISPR-Cas system [104]. CRISPR-armed phages can target plasmids or genomic regions that confer phage resistance, ensuring that bacteria remain susceptible to phage attacks over time [105]. Alternative methods exist to address the risk of developing phage resistance, such as pre-adapting phages by choosing mutants with enhanced infectivity and a lower tendency to produce phage-resistant bacterial variants. This approach demonstrated success in treating a drug-resistant *Klebsiella pneumoniae* infection [104,106].

Multiple studies have shown the therapeutic potential of employing CRISPR-Cas systems from different bacteria, e.g., *Streptococcus pyogenes*, *Staphylococcus epidermidis*, *Streptococcus thermophilus*, and *Listeria monocytogenes*, in phage engineering [68]. Among these, the CRISPR-Cas system from *S. pyogenes* is the most widely utilized in phage bioengineering due to its high efficiency in recognizing and precisely modifying a broad spectrum of phage DNA [68]. This capability allows for the creation of recombinant phage variants with enhanced functionalities [68].

### 5.3. Phages Armed with Enzymes

Genetically engineered enzyme-armed phages have shown great potential in combating bacterial infections, particularly by targeting biofilms and enhancing antimicrobial efficacy. Lu et al. modified phages to produce an enzyme that breaks down biofilms throughout the infection [88]. Namely, T7 phages were bioengineered to produce the enzyme dispensin B (DspB), which breaks down β-1,6-N-acetyl-D-glucosamine [88]. This is a key adhesive component essential for the creation of biofilms in *E. coli* and *Staphylococcus* species [91]. Synthetic biology of phages with enzymes also includes phage lytic proteins, which are used mainly against Gram-positive bacteria [97]. A recent study showed that the use of T7Select lytic phages integrated with lytic AMPs CRAMP (cathelicidin-related antimicrobial peptides) and melittin prevented *E. coli* strains from becoming resistant to phages and AMPs [107].

In another example, the engineered phage PSA ΔLCR ply511 was designed to enhance antibacterial activity against *Listeria* species by incorporating Ply511, a lytic enzyme capable of degrading bacterial cell walls [108]. This modification allowed the phage to effectively target both phage-sensitive and phage-resistant bacteria via the activity of the incorporated lytic enzyme, broadening its host range [108]. The engineered phage demonstrated strong lytic activity against multiple *Listeria* strains, including *Listeria monocytogenes* F2365 and EGDe [108].

In a recent study, a nanozyme-functionalized phage system was developed, where phages were modified with palladium nanozymes to enhance their antibacterial properties [109]. The engineered phages leveraged the enzyme-like activity of the nanozymes to generate toxic hydroxyl radicals in infection sites, effectively eliminating bacteria [109]. This strategy demonstrated significant potential in treating bacterial pneumonia and subcutaneous abscesses while assisting tissue healing [109]. However, it is important to note that while the phage component provides bacterial specificity, the hydroxyl radicals produced by the nanozymes exert a non-specific oxidative effect. This trade-off may reduce the precision traditionally associated with phage therapy but could be advantageous in contexts where broad antibacterial activity is needed, such as in treating mixed infections or disrupting biofilms.

### 5.4. Phages Armed with Nanoparticles

Metal NPs are nano-scale materials composed of a single metallic element or its oxide [110]. NPs serve as carriers that boost the delivery of antimicrobial substances and act as an inventive antimicrobial material, different from traditional medications [110]. NP-based antimicrobials work against bacteria mainly by disrupting the membrane or by causing oxidative stress via ROS production, aided by their catalytic properties [98]. Their tiny size gives them a large surface area, boosting reactivity and efficiency in killing bacteria by penetrating and damaging internal cell elements [96]. Furthermore, NPs can deliver drugs or genes to desired bacterial targets, overcoming issues like poor antibiotic penetration, low bioavailability, and resistance [96].

Phages may lose their antibiofilm activity over time, resulting in the re-establishment of biofilm to its pre-phage state [90]. Therefore, it is crucial to assess the effectiveness of the biomaterial over an extended period [90]. T7 phages are highly versatile, as they can be readily modified to display peptides or proteins that precisely interact with designated targets, ensuring strong and specific binding [111]. T7 phages and engineered T7Ag-XII phages conjugated with lower concentrations of silver nanoparticles (AgNPs) against *E. coli* have been shown to markedly enhance biofilm elimination, especially after 48 h, a period during which biofilms generally develop resistance [90]. The T7Ag-XII-AgNPs biomaterial demonstrated notably greater effectiveness over an extended duration compared to the usage of unmodified phages or nanomaterials independently [90].

Another example of an NP-armed phage is the photocatalytic quantum dot-armed phage (QD@Phage) developed by Wang et al. [89]. This “armed phage” is designed to target green fluorescent protein (GFP)-expressing *Pseudomonas aeruginosa* that generates ROS and exhibits an enhanced ability to kill bacteria and combat biofilm formation [89]. QD@Phage demonstrated significant success in eradicating the infection and assisting wound healing in a mouse model [89].

### 5.5. Phages Armed with Photosensitizers

Therapy with phages armed with photosensitizers has become a potential alternative solution for the effective destruction of MDR bacteria. Once phages successfully infect the targeted bacteria, the associated photosensitizers have the potential to damage the bacterial biofilm and the biofilm-covered bacteria upon activation with visible light [91]. Bacterial keratitis is mostly caused by MDR *Pseudomonas aeruginosa*. Chen et al. modified phages enhanced with the type I photosensitizer (PS) ACR-DMT, creating an innovative eyedrop treatment that merges phage therapy with photodynamic therapy (PDT) to combat *P. aeruginosa*-induced bacterial keratitis [91]. The therapeutic potential of the ACR-DMT-armed phage was explored on a clinical MDR-PA strain from a patient with dacryocystitis [91]. The armed-phage eyedrops enhanced bacterial clearance by reducing oxygen reliance, while perforating and breaking down biofilms and intercepting their regrowth [91]. Overall, ACR-DMT-armed phage demonstrated increased biosecurity and the potential to minimize vision impairment, thus offering a novel therapeutic alternative to conventional antibiotics [91].

Another group developed an advanced phage-based therapy that integrates an aggregation-induced emission photosensitizer (AIE-PS) with phages to enhance bacterial elimination in sepsis treatment [92]. By harnessing the AIE-PS component to generate ROS upon activation, a significant enhancement in bactericidal effects was demonstrated [92]. By combining phage cocktail therapy with PDT, this strategy significantly improved bacterial clearance, enhanced sterilization efficiency, thus demonstrating strong potential for clinical application in combating MDR infections [92].

Sioud and Zhang have developed a novel cancer treatment using “armed phages” conjugated with photosensitizers to selectively target tumor-supporting immune cells [112]. These modified phages bind specifically to M2 macrophages and, upon exposure to near-infrared light, activate the IR700 photosensitizer, leading to the destruction of these immunosuppressive cells while sparing M1 macrophages [112]. Incorporating tumor-specific peptides allows the therapy to directly attack cancer cells, creating a dual-action approach that enhances immunotherapy and disrupts the tumor microenvironment [112].

There are several research groups around the world focusing on “armed phage” research, particularly for applications like phage therapy, diagnostics, and combating antibiotic resistance. They are listed in Table 2.

## 6. Challenges and Risks Associated with “Armed Phage” Applications

Although “armed phages” represent a promising advancement in therapeutic development, their use presents several regulatory, ecological, and biosafety challenges that warrant careful consideration. The regulatory landscape for phage therapy differs substantially between Western countries and parts of Eastern Europe, creating challenges for global standardization [124]. In the United States and the European Union, phage therapy lacks a clear and unified regulatory framework and is typically authorized only on a case-by-case basis for patients with life-threatening infections that do not respond to conventional therapies [125]. However, some countries within the EU, such as Belgium, have established national frameworks, like the magistral preparation approach, that facilitate personalized phage therapies under regulated pharmaceutical standards [125]. This ambiguity poses barriers to large-scale clinical trials and commercialization. In contrast, countries such as Georgia and Russia have adopted more permissive approaches, with Georgia notably allowing the clinical use of both pre-formulated and personalized phage preparations [126]. The introduction of “armed” phages, which may involve genetic modification or synthetic biology techniques, adds further complexity and is likely to face even stricter scrutiny, potentially delaying their clinical implementation.

Another important limitation is the lack of comprehensive pharmacokinetic and pharmacodynamic (PK/PD) data for phages, which complicates the establishment of optimal dosing strategies [127]. Due to the self-replicating nature of phages at infection sites and their distinct behavior compared to antibiotics, more clinical studies are needed to fully characterize their PK/PD profiles and guide therapeutic use [127].

An additional consideration is that “armed” phages often remain reliant on the biological characteristics of their wild-type progenitors. For example, CRISPR-Cas-equipped phages may still exhibit a narrow host range or remain vulnerable to bacterial resistance, limiting their long-term efficacy. In addition, the ecological consequences of releasing engineered or synthetic viruses into natural microbial communities remain poorly understood [128]. Unintended horizontal gene transfer, disruption of microbial balance, or unforeseen interactions with environmental bacteria could theoretically pose ecological risks [128]. It is important to study the phage dynamics in nature, using examples from the lab, so that we can develop strategies to monitor them safely [129]. Based on this knowledge, appropriate biosafety and containment strategies can be standardized to minimize the risk of accidental release or misuse.

More advances in synthetic biology offer opportunities to enhance the design and performance of phage-based therapeutics. For example, tailocins, non-replicating phage tail-like particles, retain high target specificity while eliminating the risks associated with phage replication [130]. In addition, engineered lysins incorporating multiple muralytic domains, such as glycosidases, amidases, and endopeptidases, also represent a promising direction for enhancing antibacterial potency [131]. These tools may help expand the therapeutic utility of phage-derived agents while enabling more controlled and targeted interventions.

These considerations underscore the need for interdisciplinary collaboration among scientists, regulators, and public health authorities to ensure the responsible advancement of phage engineering technologies. One suggestion that could circumvent this issue and help monitor the release of engineered phages (and phages in general) is the formation of a Phage Foundry Framework, which aims to characterize fully novel phages and their interaction with their host and observe their action from an ecological perspective. The Phage Foundry Framework aims to develop a hub that brings together phage scientists from all disciplines and stakeholders to offer quality control on relevant phage-based applications and phage-involved therapies and their consequences [132].

## 7. Clinical Trials of Genetically Engineered Phages

Gencay et al. combined four type I-E CRISPR-Cas-armed phages (CAPs) from a screened library of phages and created the phage cocktail SNIPR001 to target *E. coli* in mice and minipigs as model organisms [123]. A library with 162 lytic phages was initially created and assessed against a panel of 82 representative *E. coli* strains [123]. The treatment consists of a cocktail of four selected phages armed with CRISPR-Cas systems to enhance bacterial elimination while minimizing resistance development [104]. Preclinical studies in mice and pigs thus demonstrated that SNIPR001 effectively reduced *E. coli* levels without triggering significant immune responses or disrupting the gut microbiota [104]. Although complete bacterial clearance was not achieved, no bacteria developed resistance to all four phages, highlighting the therapy’s potential advantage over traditional phage treatments [104]. The reason for this might stem from some partial resistance developed as one isolate was found resistant to one of the four phages in this cocktail. Alternatively, some bacteria may have evolved other counter-mechanisms, e.g., mutations in phage receptors. This unintended effect could be overcome by increasing the number of phages in SNIPR001 and engineering CAPs with additional CRISPR payloads targeting known *E. coli* resistance genes. A Phase I clinical trial (NCT05277350) to evaluate safety in healthy subjects has been completed, but the data have not yet been published [133]. If successful in further trials, this approach may lead to the development of similarly engineered phage therapies for other drug-resistant infections.

A new clinical trial (ELIMINATE) is evaluating LBP-EC01, a genetically engineered bacteriophage cocktail designed to treat *E. coli*-induced urinary tract infections (UTIs), particularly in patients with a history of drug-resistant infections [134]. The first phase of this ongoing trial aimed to determine an optimal dosing regimen by administering LBP-EC01 both intraurethrally and intravenously, alongside the antibiotic trimethoprim–sulfamethoxazole (TMP-SMX) [134]. LBP-EC01 was well tolerated, with no serious adverse effects, though mild reactions were more frequent at higher intravenous doses [134]. Importantly, the treatment led to a rapid and sustained reduction in *E. coli* levels in urine, with all evaluable patients experiencing complete symptom resolution by day 10 [134]. Now in Phase II, this randomized, double-blind, active-controlled trial is set to further assess the safety, tolerability, pharmacokinetics, and efficacy of LBP-EC01, potentially establishing it as an alternative to antibiotics for treating drug-resistant UTIs [135].

Table 3 provides an overview of engineered phages that have progressed to clinical trials.

## 8. Conclusions

Bacterial defense systems present significant barriers to the widespread clinical application of phage therapy, necessitating innovative solutions to enhance its efficacy. While natural phages can experience difficulties in proceeding with their infection cycles when faced with bacterial strategies such as adsorption blocking, restriction–modification, CRISPR-Cas immunity, abortive infection, and superinfection exclusion, advances in phage engineering offer promising countermeasures. By modifying receptor-binding proteins, integrating anti-CRISPR genes, incorporating methylation systems, and leveraging depolymerases or tailored lytic enzymes, “armed phages” have the potential to overcome bacterial resistance and expand therapeutic options. Beyond these targeted modifications, the development of “armed phages”—modified to deliver antimicrobial peptides, CRISPR payloads, or immune-modulating molecules—represents the next generation of phage therapy. Early clinical trials of engineered phages, such as SNIPR001 and LBP-EC01, underscore their therapeutic potential in human infections, setting the stage for future translational research. As research progresses, the continued refinement of phage modification techniques, regulatory advancements, and further clinical trials will be essential to fully realize the potential of “armed phages” in combating MDR bacterial infections. The integration of synthetic biology with phage therapy represents a promising frontier in the development of next-generation antimicrobial strategies.

## Figures and Tables

**Figure 1 viruses-17-00911-f001:**
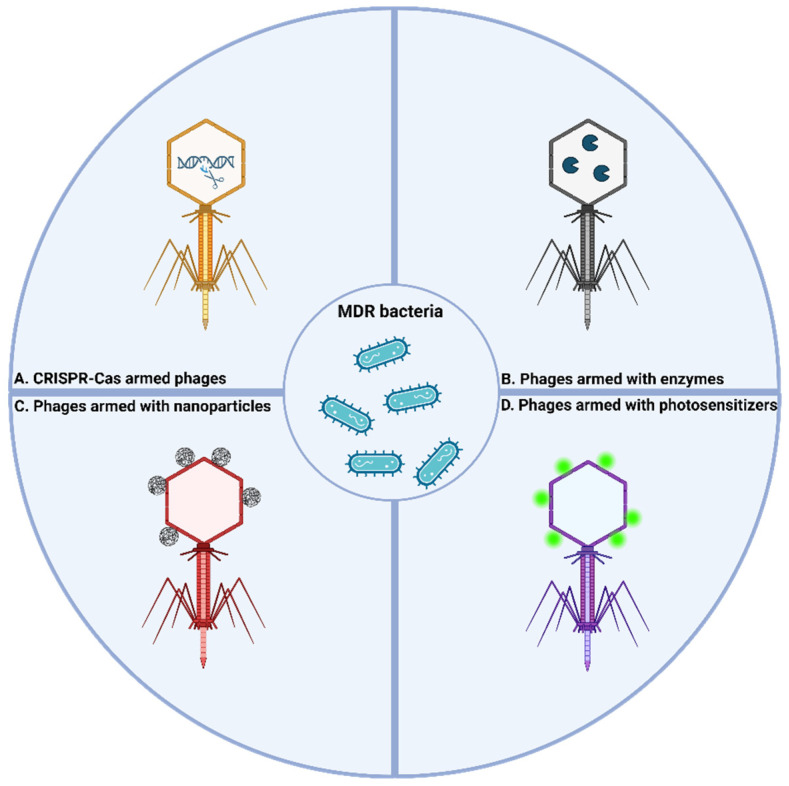
Types of armed phages for enhanced therapy.

**Table 1 viruses-17-00911-t001:** Bacterial defense challenges in phage therapy and bioengineered solutions.

Bacterial Defense Mechanism		Limitations on Phage Therapy	Solutions Offered by Bioengineered Phages
Adsorption Blocking	Surface receptor modifications	Block binding and infection of bacterial hosts	Modify phage to recognize multiple bacterial receptors, expanding host range
	Extracellular barriers	Prevent phages from reaching specific receptors on bacteria	Engineer phages to produce depolymerase enzymes that degrade bacterial polysaccharides
Nucleic acid protection	R-M	Bacterial restriction enzymes degrade phage DNA	Introduce methylation systems to protect phage DNA
	CRISPR-Cas	Recognize and cleave phage DNA	Engineer phages to carry Acr genes that inhibit CRISPR defenses
Abortive infection		Premature bacterial cell death before phage replication	Engineer phages to express proteins to suppress the Abi system effects
NAD+ depletion systems		NAD⁺ degradation disrupts phage replication and induces bacterial cell death	Engineer phages to express NAD⁺ reconstitution pathways

NASD^+^, nicotinamide adenine dinucleotide⁺; R-M, restriction–modification.

**Table 2 viruses-17-00911-t002:** Research groups specializing in bioengineering armed phages for therapeutic or diagnostic applications.

Scientific Groups	Affiliation	Research Focus	Key References
Briers et al.	Department of Biotechnology, Ghent University, Ghent, Belgium	Engineering phages	[113,114]
Golec et al.	Department of Molecular Virology, University of Warsaw, Poland	Phages armed with nanoparticles	[90,110]
Hatfull et al.	Department of Biological Sciences, University of Pittsburgh, USA	Mycobacteriophages	[115,116]
Loessner et al.	Institute of Food, Nutrition and Health, ETH, Zürich, Switzerland	Engineering bacteriophages for therapeutic and diagnostic purposes	[117,118]
Qimron et al.	Department of Clinical Microbiology and Immunology, School of Medicine, Tel Aviv University, Israel	Engineering phages	[100,119]
Sagona et al.	Department of School of Life Sciences, University of Warwick, UK	Engineering phages for therapeutic and diagnostic purposes	[120,121,122]
Sommer et al.	SNIPR BIOME ApS, Copenhagen, Denmark	CRISPR-Cas armed phages	[123]

**Table 3 viruses-17-00911-t003:** Clinical trials for bioengineered phage therapies.

Identifier	Target Bacteria	Engineering Strategy	Study Design	Study Details	Endpoints
SNIPR001 NCT05277350 [104,133]	*E. coli* in the gut of high-risk hematological cancer patients	Engineered with tail fibers and CRISPR-Cas machinery Cocktail of 4 phages	Phase I, randomized, double-blind, placebo-controlled	Ascending oral doses (BID × 7 days) 36 healthy participants randomized to 3 cohorts Follow-up over 6 months post-treatment	Primary: Safety and tolerability through day 35 Secondary: AEs through day 187 and PD (functional quantification of study drug in blood, feces, and urine)
LBP-EC01 NCT05488340 [134,135]	*E. coli* in acute uncomplicated UTI	Recombinant phages Cocktail of 6 phages	Phase II, randomized, double-blind, placebo-antibiotic-controlled	Multiple oral doses (BID × 3 days) 288 patients, blinded 1:1 randomized trial	Part 1: Dose regimen selection—open-label, 30 patients, 3-arm PK assessment Part 2: Efficacy, safety, tolerability, and PK

AEs, adverse events; BID, twice a day; PD, pharmacodynamics; PK, pharmacokinetics; UTI, urinary tract infection.

## Data Availability

Not applicable.

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
