# Peer review of "Armed Phages: A New Weapon in the Battle Against Antimicrobial Resistance"

_viruses, 2025, doi:10.3390/v17070911_

Round 1
Reviewer 1 Report
Comments and Suggestions for Authors This review explores the emerging role of “armed phages” as a novel antibacterial tool. Covering bacterial defenses, phage-engineering strategies to overcome them, and clinical trials of armed phages, the article highlights their potential in addressing antibiotic resistance. However, room for improvement remains in terms of content depth and detail.Specific Comments
- Page 2: When introducing the lytic and lysogenic cycles of phages, it would be helpful to include specific examples to better illustrate the characteristics and differences between these two cycles.
- Page 3: The section on phage infection mechanisms, particularly "phage hijacks the host’s molecular machinery for replication...", is somewhat brief. A more detailed explanation of how phages utilize the host’s ribosomes, enzymes, and energy systems for transcription, translation, and replication would be beneficial. Additionally, information on the functions and regulatory mechanisms of different gene expression stages would add depth.
- Page 5: In the discussion of the CRISPR-Cas system, the segment "phages have developed multiple strategies to evade bacterial CRISPR-Cas defenses..." could be expanded. Specifically, the mechanisms by which various anti-CRISPR proteins interact with and inhibit Cas proteins should be elaborated, along with differences between anti-CRISPR protein families.
- Page 6: The description of the NAD+ depletion defense system, particularly "TIR domains respond to phage infection by synthesizing nicotinamide (NAM)...", lacks sufficient detail on how TIR domains detect phage infections and activate downstream signaling pathways. Relevant research findings could be incorporated here.
- Page 8: In the section on genetically engineering phages to enhance therapeutic potential, specifically "modification of phage receptor-binding proteins...", more research case studies would strengthen the discussion. For instance, explaining how modifying phage receptor-binding proteins expands host ranges and impacts infection efficiency and therapeutic outcomes would be valuable.
- Page 11: The discussion on challenges and risks of armed phage applications, particularly "the ecological consequences of releasing engineered or synthetic viruses into natural microbial communities remain poorly understood...", could be enhanced by exploring ecological risk assessment and monitoring strategies. This would help demonstrate how to mitigate potential environmental impacts and balance therapeutic benefits with ecological risks.
Reviewer 2 Report
Comments and Suggestions for Authors
The review article covers the wide concepts of phages, which are concerning and confusing. For better understanding, the authors should write about phage engineering and their importance in therapy.
- Pg.2: There is no evidence (from the clinical trials) to prove that the bacterial defense system is directly involved in phage therapy failure.
- Pg.2: Phage replication is misdescribed. There are early, middle and late genes in most cases.
- Bacterial defense strategies have not been explained from the therapeutic point of view.
- There are lots of other alternative strategies being employed to improve phage therapy, such as phage cocktails, phage-antibiotic combinations.
- Use of natural phages has therapeutic approval on a case-by-case basis, but the use of engineered phages for therapy is not approved yet. Drug delivery using phage weapons is not considered phage therapy.
Reviewer 3 Report
Comments and Suggestions for Authors
Comments to the manuscript viruses-3658170 titled “Armed phages: a new weapon in the battle against antimicrobial resistance”
Phage therapy is one of most promising alternatives for combating MDR pathogens. Bacteriophages were among the first antibacterial agents discovered to be antibacterial agents and were used in the former Soviet Republic of Georgia (pioneered there by Giorgi Eliava with help from the co-discoverer of bacteriophages, Félix d'Hérelle) during the 1920s and 1930s for treating bacterial infections, but was not widely adopted due to the rapid and widespread use of penicillin. Given that the world is currently facing the problem of the emergence of new and rapidly spreading antibacterial resistance, alternatives to antibiotics are being sought so that we do not enter the post-antibiotic era unprepared. There are numerous, but not as effective, alternatives to antibiotics, which include bacteriophages, antimicrobial peptides of various origins (defensins, bacteriocins, plant APs, etc.), lipopeptides, essential oils, probiotics. Each of the antimicrobial agents has its own advantages and disadvantages; for example, the broad spectrum of antibiotic activity facilitates its use against various pathogens but negatively affects the host microbiota, which is also very important for ensuring good host health. The specificity of phages for even a unique strain of bacteria is, on the one hand, an advantage because it does not affect the microbiota and eliminates the pathogen in a targeted manner, but it is challenging to provide an effective phage for each strain of pathogen. On the other hand, bacteria, more or less quickly, develop defense mechanisms against all types of antimicrobial agents, and it is only a matter of time when mutants resistant to new antimicrobial agents will appear. Most experts believe that if a new therapeutic with a different mechanism of action were to be applied over a decade, the genes responsible for antibiotic resistance would likely be eliminated since there would be no selective pressure to maintain them, so antibiotics could be used again. Given that the use of phages as therapeutics has positive support in individual cases as a last chance for survival from MDR pathogens, but due to shortcomings that may jeopardize it, the authors in this review seek a way out in engineered “armed-phages” that would represent super-powerful agents that would always achieve results because they would be based on multiple mechanisms of action and on the other hand, they would be able to disarm the defense mechanisms of bacteria. Although there are positive examples of the superiority of engineered phages, there are definitely obstacles to their development and implementation as therapeutics at various levels such as legal, environmental, developmental, and scientific. We increasingly encounter authors who biasedly extol the importance of a new potential therapeutic by giving it unrealistic powers (even omnipotence). This review is written quite realistically with small excursions into the impossible. Perhaps the language should be improved and some references should be changed to more adequate ones. Given that AMR is a real threat to humanity, especially to immunocompromised individuals, we must take a step forward, no matter what the cost, to prepare for the ineffectiveness of current therapeutics, and one way to prepare is to point out the problem and consider alternatives, which should contribute to the development of solutions that require the involvement of all of humanity, not just scientists, politicians and the pharmaceutical industry.
Minor suggestions:
Title “Armed phages: a new weapon in the battle against antimicrobial resistance”
I am not aware of how applicable armed phages are against microorganisms other than bacteria. Therefore, I suggest that the title focus on antibacterial (or antibiotic) resistance.
Abstract: The increasing prevalence of multidrug-resistant (MDR) bacterial infections necessitates
the exploration of alternative antimicrobial strategies, with phage therapy emerging as a viable option.
The increasing prevalence of multidrug-resistant (MDR) bacterial infections requires the exploration/research/investigation of alternative antibacterial strategies, with phage therapy emerging as a viable/promising option.
The use of antibiotics revolutionized medicine, drastically improving patient outcomes
and saving millions of lives [1].
Maybe that better citation is:
Park M (2023) Importance of Antibiotics in Revolutionizing Medicine and Challenged by Resistance. Adv Tech Biol Med. 11:410.
However, the widespread misuse and overprescription of these drugs created intense selective pressure on bacteria, fostering the emergence of multidrug-resistant (MDR) strains that no longer respond to conventional treatments [2].
Maybe that better citations are:
Davies J, Davies D. Origins and evolution of antibiotic resistance. Microbiol Mol Biol Rev. 2010 Sep;74(3):417-33. doi: 10.1128/MMBR.00016-10. PMID: 20805405; PMCID: PMC2937522.
Habboush Y, Guzman N. Antibiotic Resistance. [Updated 2023 Jun 20]. In: StatPearls [Internet]. Treasure Island (FL): StatPearls Publishing; 2025 Jan-. Available from: https://www.ncbi.nlm.nih.gov/books/NBK513277/
As a result, antimicrobial resistance (AMR) has become a major global health crisis.
Maybe that better term is “issue/problem” instead “crisis”?
Future projections indicate an even more alarming scenario, with an estimated 1.91 million deaths directly attributed to drug-resistant infections and 8.22 million AMR-related deaths anticipated by 2050 [4].
I think that citation (de Kraker et al. 2016) should be included?
de Kraker ME, Stewardson AJ, Harbarth S. Will 10 Million People Die a Year due to Antimicrobial Resistance by 2050? PLoS Med. 2016 Nov 29;13(11):e1002184. doi: 10.1371/journal.pmed.1002184. PMID: 27898664; PMCID: PMC5127510.
The rise of AMR, coupled with the lack of discovery of new antibiotics in over three decades, has compelled the scientific community to prioritize the development of alternative treatments to combat MDR bacteria [5].
Maybe that citation (Martins & McCusker, 2016) could be included?
Martins M, McCusker MP. Editorial: Alternative Therapeutics against MDR Bacteria - "Fighting the Epidemic of Antibiotic Resistance". Front Microbiol. 2016 Oct 7;7:1559. doi: 10.3389/fmicb.2016.01559. PMID: 27774086; PMCID: PMC5053995.
One of these alternative treatments involves the resurgence of phage therapy, nearly 100 years after it was abandoned in Europe and the United States [2].
Gordillo Altamirano FL, Barr JJ. Phage Therapy in the Postantibiotic Era. Clin Microbiol Rev. 2019 Jan 16;32(2):e00066-18. doi: 10.1128/CMR.00066-18. PMID: 30651225; PMCID: PMC6431132.
In 2022, the International Commi􀄴ee on Taxonomy of Viruses (ICTV) formally abolished the morphology-based families Myoviridae, Siphoviridae and Podoviridae, along with the order Caudovirales, due to their polyphyletic nature [7].
Valencia-Toxqui G, Ramsey J. How to introduce a new bacteriophage on the block: a short guide to phage classification. J Virol. 2024 Oct 22;98(10):e0182123. doi: 10.1128/jvi.01821-23. Epub 2024 Sep 12. PMID: 39264154; PMCID: PMC11494874.
The design of their capsids varies significantly, influencing the mode of infection of host bacteria.
Naureen Z, Dautaj A, Anpilogov K, Camilleri G, Dhuli K, Tanzi B, Maltese PE, Cristofoli F, De Antoni L, Beccari T, Dundar M, Bertelli M. Bacteriophages presence in nature and their role in the natural selection of bacterial populations. Acta Biomed. 2020 Nov 9;91(13-S):e2020024. doi: 10.23750/abm.v91i13-S.10819. PMID: 33170167; PMCID: PMC8023132.
The lysogenic cycle involves the integration of the phage genome into the bacterial host's DNA, allowing it to remain dormant and replicate with each bacterial cell division [11].
Roughgarden J. Lytic/Lysogenic Transition as a Life-History Switch. Virus Evol. 2024 Apr 3;10(1):veae028. doi: 10.1093/ve/veae028. PMID: 38756985; PMCID: PMC11097211.
The rise of antibiotic resistance has created an urgent need for alternative antimicrobial strategies, with phage therapy emerging as a promising solution for combating MDR bacterial infections.
Kakasis A, Panitsa G. Bacteriophage therapy as an alternative treatment for human infections. A comprehensive review. Int J Antimicrob Agents. 2019 Jan;53(1):16-21. doi: 10.1016/j.ijantimicag.2018.09.004. Epub 2018 Sep 17. PMID: 30236954.
The effectiveness of phage therapy may nonetheless be hindered by bacterial defense systems, which have evolved to block phages’ infection through diverse molecular mechanisms. These barriers highlight a critical challenge in the clinical application of natural phages.
The host/human immune response to phages particles can be also limiting factor for their application (discussed in Kakasis and Panitsa, 2019). and also in:
Berkson JD, Wate CE, Allen GB, Schubert AM, Dunbar KE, Coryell MP, Sava RL, Gao Y, Hastie JL, Smith EM, Kenneally CR, Zimmermann SK, Carlson PE Jr. Phage-specific immunity impairs efficacy of bacteriophage targeting Vancomycin Resistant Enterococcus in a murine model. Nat Commun. 2024 Apr 6;15(1):2993. doi: 10.1038/s41467-024-47192-w. PMID: 38582763; PMCID: PMC10998888.
Phage infection begins with adsorption, where phages a􀄴ach to bacterial hosts via interactions between viral receptor-binding proteins and bacterial surface receptors [13].
This attachment is achieved through specific interactions between phage receptor-binding proteins (RBPs) and bacterial surface receptors.
Bloch S and Wegrzyn A (2024) Editorial: Bacteriophage and host interactions. Front. Microbiol. 15:1422076. doi: 10.3389/fmicb.2024.1422076
Depolymerases can aid adsorption by degrading capsular polysaccharides that block bacterial receptors [14]. After a􀄴achment, virion-associated lysins (ectolysins) degrade peptidoglycan locally, creating an entry point for the phage genome [14].
Bacteriophage-derived enzymes depolymerases can aid/facilitate phage adsorption by degrading capsular polysaccharides that block/mask receptors on bacterial surface. After attaching to a bacterial cell, phage-encoded enzymes called virion-associated lysins (also known as ectolysins) locally degrade the bacterial peptidoglycan layer, creating a pathway for the phage genome to enter the host cell. This local peptidoglycan degradation is crucial for initiating the infection process.
Early genes modify host transcription and aid replication, while later genes encode structural proteins for virion assembly and host lysis [17,18].
In viruses, early genes are expressed first and primarily control host cell modifications and viral DNA replication, while later genes are then expressed to encode structural proteins for building the viral virion (the external infectious particle) and proteins involved in host cell lysis. This sequential gene expression is crucial for the viral life cycle.
Bacterial defense strategies against phages
Understanding the mechanisms by which phages infect and hijack bacterial cells pro-
Maybe that term “subjugate” is better than “hijack”?
These bacterial defense systems, ranging from surface modifications that prevent phage adsorption to sophisticated intracellular immunity pathways, reflect the ongoing evolutionary arms race between bacteria and phages [21].
The ongoing coevolution between bacteria and bacteriophages is a dynamic process where both organisms constantly evolve in response to each other, leading to the development of resistance and counter-resistance strategies.
Nevertheless, conjugative plasmids can acquire Acr proteins, which disable CRISPR defenses and allow antibiotic resistance genes to evade destruction and spread between bacteria [67].
I do not understand this sentence; What does this have to do with bacteriophages and their use as therapeutics?
“Armed phages” phages have been shown to boost the……..
It is not necessary to be two times “phages”?
The regulatory landscape for phage therapy differs substantially between Western countries and parts of Eastern Europe, creating challenges for global standardization.
Strathdee, Steffanie, Patterson, Thomas L., & Barker, Teresa. (2019). The perfect predator: A scientist's race to save her husband from a deadly superbug. Hachette Books.
Comments on the Quality of English Language
I am not qualified to judge English quality
Reviewer 4 Report
Comments and Suggestions for Authors
This review explores the primary defense mechanisms employed by bacteria and their implications for phage therapy. It highlights how phage bioengineering is being innovatively utilized to tackle these obstacles as "armed phages" advance into clinical use. Through synthetic biology, phages can be genetically modified to bypass bacterial defenses via strategies such as redesigning receptor-binding proteins, integrating anti-CRISPR genes, adjusting methylation patterns, and deploying enzymes to break down bacterial protective layers. "Armed phages," equipped with antimicrobial peptides, CRISPR-based editing systems, or immune-modulating agents, present a groundbreaking therapeutic approach. Early-stage clinical trials, including SNIPR001 and LBP-EC01, demonstrate the safety and efficacy of these engineered phages in treating multidrug-resistant (MDR) infections. As "armed phages" move closer to real-world medical applications, they offer significant potential for precision-targeted treatments and stand as a pivotal advancement in combating the global antibiotic resistance crisis. The fusion of synthetic biology with phage therapy marks a promising direction for next-generation antimicrobial solutions.
I believe that this review can be published after a significant change in the text. Individual passages in the text are very difficult to understand. If you read the articles referred to by the authors, then their idea becomes clear. I have read many such articles, and the question arises about the need for such a review?
Comments:
- “One of these alternative treatments involves the resurgence of phage therapy, nearly 100 years after it was abandoned in Europe and the United States [2].”
However, in Georgia (Europe) and Poland (Europe), such therapy continued to be used throughout the 20th century.
- “A potential solution discussed during the Centennial Celebration of Bacteriophage Research in Georgia in 2017 was the use of phage cocktails [57].”
It looks like phage cocktails were invented in Georgia in 2017. However, they have been known for a very long time. It was used in the early 20th century.
- Another important category includes phages armed with enzymes, which produce cell wall degrading enzymes capable of disrupting bacterial biofilms [72].
enzymes, which produce cell wall degrading enzymes?
- CRISPR-Cas systems include types II-A (CRISPR-Cas9), I-E (CRISPR-Cas3), III-A (CRISPR-Cas10), V (CRISPR-Cas12a), and VI-A (CRISPR-Cas13a) [75].
and many others: I-A, I-B, I-C, I-D, …… VI-X
CRISPR-Cas9 is the most commonly used system, while the type I CRISPR-Cas complex includes Cas3 and the CRISPR-associated complex for antiviral defense [82].
What is the relationship between Cs3 and Cas9?
The CRISPR-Cas9 system can combat methicillin-resistant Staphylococcus aureus (MRSA) infections in vivo [83].
It's about Cas9 again
Kiro et al. were the first to genetically engineer the Escherichia coli phage T7 with the application of type I-E CRISPR-Cas system [84].
Now about Cas3. It is unclear what the authors want to say.
Using homologous recombination, the T7 phage genome was modified, and the CRISPR-Cas system targeted unedited genomes [84]. Subsequently, the intended recombinant phages were separated, rendering this CRISPR-Cas system applicable for modifying any phage genome beyond the type II CRISPR-Cas, which incorporates the versatile Cas9 protein [84].
It is unclear what the authors want to say.
CRISPR-Cas3 was successful in targeting Clostridium difficile both in vivo and in vitro [85].
Other variants, such as CRISPR-Cas12, specialize in targeting DNA, whereas CRISPR-Cas13 is designed to recognize RNA, making it a powerful tool for controlling gene expression of RNA viruses [82]. CRISPR-Cas13 can effectively target antimicrobialresistant E. coli and S. aureus strains [86].
It's not true! CRISPR is a family of DNA sequences, Cas is an enzyme. If they are placed in an environment with E. coli, they will not enter bacteria and will not be a powerful tool for controlling the expression of RNA virus genes.
CRISPR-Cas13 offers powerful antiviral protection
(Whose powerful antiviral protection does CRISPR-Cas 13 provide?)
and precise, widespread phage genome modification [87].
(any phage? How accurate is it?)
- T7 phages were bioengineered to produce the enzyme dispensing B
Change to dispensing
- This modification allowed the phage to effectively target both phage-sensitive and phage-resistant bystander cells
What do the authors mean by bystander cells?
- Multiple studies have shown the therapeutic potential of genetically manipulating the CRISPR-Cas system from different bacteria
What does it mean? Have the researchers altered CRISPR-Cas in different bacteria?
- In the United States and the European Union, phage therapy lacks a clear and unified regulatory framework and is typically authorized only on a case-bycase basis for patients with life-threatening infections that do not respond to conventional therapies [108].
As far as I know, the regulatory situation in EU countries such as Poland and Belgium is completely different from that described by the authors.
- Gencay et al. combined four type I-E CRISPR-Cas-armed phages (CAPs) and created the phage cocktail SNIPR001 to target E. coli in mice and minipigs [107].
So this cocktail is specific to mice and mini-pigs and won't be effective, for example, for dogs?
A library with 162 lytic phages was created and assessed against a panel of 82 representative E. coli strains [107].
I didn't understand, did the authors write earlier about 4 phages?
The treatment consists of a cocktail of four phages armed with CRISPR-Cas systems to enhance bacterial elimination while minimizing resistance development [88].
4 phages appeared again
Preclinical studies in mice and pigs demonstrated that SNIPR001 effectively reduced E. coli levels without triggering significant immune responses or disrupting gut microbiota [88].
Previously, "the authors created the phage cocktail SNIPR001 to target E. coli in mice and minipigs," i.e. are these clinical trials?
Reviewer 5 Report
Comments and Suggestions for Authors
In this review author discussed the bacterial defense mechanisms and their implications for phage therapy, while highlighting the potential of phages’ bioengineering to overcome these barriers. The review is written very well and in the following section author will find few comments for their consideration.
Comments:
- Page 2, last paragraph: Author mentioned “The rise of antibiotic resistance has created an urgent need for alternative antimicrobial…………” the similar lines were used on page 1-2, first paragraph.
- Page 2, section 2: If author could show the evolutionary dynamics of phage and bacterial interaction by a pictorial cartoon that would be great for the clear understanding of the readers.
- Page 8, section 5: Author can also mention about the phage and antibiotic combination for the eradication of the MDR infections.
- Page 12, section 6: Author could also discuss about the “poor phage PK knowledge” which could affect the phage dosing in human.
Round 2
Reviewer 2 Report
Comments and Suggestions for Authors
The author's response is satisfactory, and the required changes have been made to the manuscript. Interested in the two (missing) topics: 1) engineering phage particles as tailocins, and 2) in the enzyme-armed phages section, how about engineering multiple muralytic enzymes (commonly found in phages are glycosidase, transglycosylase, amidase and endopeptidase)
Reviewer 3 Report
Comments and Suggestions for Authors
Authors improved manuscript accordingly. They accept all suggestions of reviewers.
Reviewer 4 Report
Comments and Suggestions for Authors
This review explores the primary defense mechanisms employed by bacteria and their implications for phage therapy. It highlights how phage bioengineering is being innovatively utilized to tackle these obstacles as "armed phages" advance into clinical use. Through synthetic biology, phages can be genetically modified to bypass bacterial defenses via strategies such as redesigning receptor-binding proteins, integrating anti-CRISPR genes, adjusting methylation patterns, and deploying enzymes to break down bacterial protective layers. "Armed phages," equipped with antimicrobial peptides, CRISPR-based editing systems, or immune-modulating agents, present a groundbreaking therapeutic approach. Early-stage clinical trials, including SNIPR001 and LBP-EC01, demonstrate the safety and efficacy of these engineered phages in treating multidrug-resistant (MDR) infections. As "armed phages" move closer to real-world medical applications, they offer significant potential for precision-targeted treatments and stand as a pivotal advancement in combating the global antibiotic resistance crisis. The fusion of synthetic biology with phage therapy marks a promising direction for next-generation antimicrobial solutions.
The authors have done a great job of editing the manuscript according to the reviewers' comments, and I believe that this manuscript can be published in Viruses.
